# Characterization of the Nero Siciliano Pig Fecal Microbiota after a Liquid Whey-Supplemented Diet

**DOI:** 10.3390/ani13040642

**Published:** 2023-02-12

**Authors:** Giuseppe Tardiolo, Orazio Romeo, Alessandro Zumbo, Marco Di Marsico, Anna Maria Sutera, Riccardo Aiese Cigliano, Andreu Paytuví, Enrico D’Alessandro

**Affiliations:** 1Department of Veterinary Sciences, University of Messina, Polo Universitario dell’Annunziata, Via Palatucci snc, 98168 Messina, Italy; 2Department of Chemical, Biological, Pharmaceutical and Environmental Sciences, University of Messina, Viale Ferdinando Stagno d’Alcontres 31, 98166 Messina, Italy; 3Sequentia Biotech SL, Carrer del Dr. Trueta 179, 08005 Barcelona, Spain

**Keywords:** Nero Siciliano pig, next-generation sequencing, 16S rRNA gene, fecal microbiota, microbial community, liquid whey supplementation, metagenomics

## Abstract

**Simple Summary:**

Liquid feeding is an alternative practice employed in swine production that allows the recovery of low-cost liquid by-products to reduce environmental impact. This type of feeding can positively affect animal’s gut health, well-being, and performance. Liquid whey is a palatable feed and it can represent a resource to administer ingredients without additional costly processes. Nowadays, the study of the gut microbiome is considered a relevant tool due to the impact of this research field on host health, well-being, and growth. Therefore, this study investigated the fecal microbiota of the autochthonous pig breed Nero Siciliano fed a liquid whey co-feed-supplemented diet using a metagenomics approach.

**Abstract:**

The utilization of dairy by-products as animal feed, especially in swine production, is a strategy to provide functional ingredients to improve gut health. This study explored the potential effect of a liquid whey-supplemented diet on the fecal microbiota of eleven pigs belonging to the Nero Siciliano breed. Five pigs were assigned to the control group and fed with a standard formulation feed, whereas six pigs were assigned to the experimental group and fed with the same feed supplemented with liquid whey. Fecal samples were collected from each individual before the experimental diet (T0), and one (T1) and two (T2) months after the beginning of the co-feed supplementation. Taxonomic analysis, based on the V3–V4 region of the bacterial 16S rRNA, showed that pig feces were populated by a complex microbial community with a remarkable abundance of *Firmicutes*, *Bacteroidetes*, and *Spirochaetes* phyla and *Prevotella*, *Lactobacillus*, *Clostridium*, and *Treponema* genera. Alpha and beta diversity values suggested that the experimental diet did not significantly affect the overall fecal microbiota diversity. However, analysis of abundance at different time points revealed significant variation in several bacterial genera, suggesting that the experimental diet potentially affected some genera of the microbial community.

## 1. Introduction

The use of by-products in animal nutrition is a well-established practice in livestock chain production to reduce the competition between human food and animal feed [1]. In addition, as antibiotic use in breeding managements has been strongly reduced or banned in several countries, we are witnessing a continuous search for possible feed to act as animal growth and health promoters [1,2]. The adoption of unconventional raw materials as alternative feedstuffs has garnered interest from feed producers [1,2]. Liquid feeding represents an alternative strategy through which breeders directly provide functional ingredients without additional costly processes [1,2]. In this view, milk co- and by-products are considered valuable feed ingredients in pigs’ diets that can be used as raw materials to support an enhanced intestinal environment [2]. Among them, liquid whey (LW), a by-product of the cheese-making process, can be considered a prebiotic feed representing a source of lactose and digestible protein that can be utilized by beneficial intestinal bacteria [3,4]. Although LW contains valuable nutrients, it is usually disposed of as a waste product, contributing to environmental pollution [5]. The exact chemical profile of LW depends on the cheesemaking processes. However, lactose is the main component, together with other common milk compounds such as mineral salts, vitamins, and soluble proteins [6,7]. Data available from the scientific literature show that whey supplementation in animals produces favorable effects on growth, immune function, and the establishment of intestinal microbiota due to its bioactive compounds such as functional amino acids, lactoferrin, and growth factors [3,4,8,9]. In pigs, its use improved a number of physiological parameters, including growth, gut health, and immunity [10,11,12]. Moreover, whey-supplemented diets were also used in the treatment of gastrointestinal disorders [13,14]. Hence, due to the paucity of data about the use of LW as alternative co-feed in autochthonous pig breed, we explored the potential effect of a liquid whey-supplemented diet on the fecal microbiota of pigs belonging to the Nero Siciliano breed.

In recent times, there has been a growing interest in the study of gut microbiota, a complex and dynamic community of microbial species, which collectively modulate the health status and physiology of a variety of vertebrates, including humans and pigs [15,16,17]. The domestic pig (*Sus scrofa domesticus*) is considered one of the most important livestock species worldwide, not only for historical, cultural, and socioeconomic reasons [18,19,20] or for the nutritional value of its meat and by-products [21,22], but also for its relevant role in many biomedical fields [23]. Pigs are monogastric omnivores with similar digestive tract anatomy, morphology, and physiology to humans [24]. This similarity was further emphasized by demonstrating that human fecal microbiomes can be transplanted into pigs, opening interesting perspectives on the generation of realistic animal models of the human gastrointestinal tract [25]. Furthermore, pigs and humans display similar susceptibilities and clinical manifestations towards several pathogens responsible for various enteric diseases [17,24,25,26]. 

The taxonomic structure of pig microbiomes is considered pivotal for commercial breeding. Additionally, the influence of microbial taxa on host fitness and disease risk can provide relevant information to promote animal health and improve feed efficiency in the swine industry [27,28]. The porcine intestinal microbiota can be considered an added organ with a crucial role in nutrient processing and managing the ingested energy [26,29,30]. Several studies have shown associations of distinct microbial profiles with nutrition and productivity parameters [31,32,33]. In particular, the gut microbiota metabolizes various food components, providing nutrients to the host as fermentation end-products and other by-products, including amino acids, vitamins, and indole derivatives [34,35].

Nero Siciliano is an autochthonous breed of black pig living in the woods of the Nebrodi and Madonie mountains, located on the northern coast of the Mediterranean island of Sicily (South Italy), well-known for its cultural and economic role due to the high quality of its meat and by-products [36]. This ancient and endangered-maintained breed is one of the six Italian autochthonous pig breeds that has been genetically well-characterized [37,38,39,40,41]. According to the traditional methods adopted in the Sicilian region, this pig is reared in extensive and semi-extensive farming systems using pasture and other natural resources for feeding. Interestingly, this breed is resistant to infectious diseases, showing great potential for adaptation to harsh environments [36]. The “Register of Native Breeds” approximately counts 13.500 animals, of which about 5.000 sows are from over 115 farms [ANAS; http://www.anas.it (accessed on 15 December 2022)]. The taxonomic composition of the Nero Siciliano fecal microbiota was determined by sequencing the V3–V4 hypervariable region of the bacterial 16S rRNA gene using a next-generation sequencing approach from feces samples collected from eleven pigs at three time points.

## 2. Materials and Methods

### 2.1. Animal Management and Experimental Design

The study involved eleven Nero Siciliano pigs indoor-reared in a suitable and authorized commercial farm located in the province of Messina (Sicily, Italy). Individuals were homogeneous for sex, body weight (average initial body weight of 19.4 ± 1.92 kg), age (58 ± 2 days), and breeding management. Animals were housed in a barn in individual pens provided with nipple waterers and stainless-steel feeders and fed individually. Five pigs were assigned to the control (CTRL) group and fed with a pellet complete feed (Table 1) rationed based on 3% of the live weight, and six pigs were assigned to the experimental (LW) group and fed with the same formulation supplemented with fresh LW at the level of 1.5 L/day/pig. LW was promptly and daily provided to the animals as fresh co-feed administered separately using a wet feeder and maintained in sterilized cans. Individuals had no gastrointestinal diseases or exposure to antibiotics before the beginning of the investigation. The study was conducted over 60 days throughout the fattening period, including an adaptation period of 10 days. All pigs were exposed to natural environmental temperatures and photoperiod. Thermal and hygrometric parameters (Appendix A) were detected using data loggers (Gemini, UK) placed inside and outside of the barn.

### 2.2. Fecal Sample Collection and Next Generation Ssequencing

Sampling was performed at different time points, starting before the experimental study (T0, May), and one (T1, June) and two (T2, July) months after the beginning of the dietary treatment. Thirty-three fecal samples were directly sampled from the rectal ampoule of the pigs. From each pig, fecal samples were manually collected using sterile plastic tubes and promptly transported to the laboratory in a dry ice-cooled container. After, an aliquot of 400 mg of each sample was immediately transferred and stored in OMNIgene^®^•GUT tubes (Voden Medical Instruments, Italy), a sterile system for fecal sample collection and stabilization of microbial DNA from feces for gut microbiome profiling. Samples were sent to Eurofins Genomics (Konstanz, Germany) for microbial genomic DNA extraction and 16S-amplicon sequencing using an Illumina MiSeq approach in 2 × 300 bp paired-end mode. 

### 2.3. Bioinformatics and Statistical Analysis

Quality checks on the Illumina raw reads were performed using the software FastQC [42]. Trimmomatic software version 0.39 [43] was used to filter out adapters and low-quality reads (Phred-score ≤ 20), filtering for a minimum read length of 50 (ILLUMINACLIP: PE. fa:2:30:10 MINLEN:50 SLIDINGWINDOW:5:20) and to trim low-quality 3′-ends of reads. Bioinformatics analysis was performed using the metagenomic pipeline “GAIA”, able to analyze next-generation sequencing data with proven high accuracy as described by Paytuví et al. [44]. Statistical analysis and data visualization were conducted in R [45]. The downstream images and plots were generated by the phyloseq package [46], whereas the vegan package was used to produce the rarefaction curve of all samples [47]. Additionally, phyloseq [46] was used to estimate the alpha diversity based on Observed species, Chao 1, and Shannon indices (Appendix A), including the beta diversity across all samples using the Bray-Curtis index [48] (Appendix A). Principal coordinate analysis (PCoA) was elaborated based on the Bray-Curtis distance [48] using the ggplot2 package [49]. The statistical tests of Wilcoxon [50] and Kruskal-Wallis [51] were calculated to estimate significant differences between the alpha diversity indices. The analysis of abundance at different time points was performed using the DESeq2 package [52]. An adjusted *p*-value of ≤ 0.05 was considered statistically significant.

## 3. Results

### 3.1. Growth Performance, Quality Check, and Taxonomic Composition of Pig Fecal Microbiota

Regarding the growth performance, the average daily weight recorded was 305 g/head per day in pigs of the control group and 310 g/head per day in pigs of the treated group. The feed conversion rate recorded was 2.80 and 2.77 kg/kg in control and treated pigs, respectively. 

Regarding the quality check, an average of 104,275 of raw reads were obtained by Illumina sequencing, whereas after the trimming step, a total of 101,252 clean reads were retained for downstream analysis. The rarefaction curve of all samples associating the sample size with the operational taxonomic units (OTUs) at genus level encountered among all fecal samples shows that an adequate sequencing deep was reached (Appendix A). Regarding the fecal microbiota profile, the V3–V4 region of the 16S rRNA was sequenced using the bacterial DNA extracted from thirty-three feces samples collected at three time points. Reads were processed and filtered using the pipeline GAIA [44], and a total of 856 OTUs at the genus level were identified among all samples. 

At the phylum level, *Firmicutes* (47%) and *Bacteroidetes* (38%) were the most abundant, whereas *Spirochaetes*, *Proteobacteria*, and *Actinobacteria* were less represented among samples (6%, 4%, and 3%, respectively). At the family level, *Prevotellaceae* (30%), *Ruminococcaceae* (14%), and *Spirochaetaceae* (9%) were the most abundant bacterial families detected. Finally, at the genus level, those most represented were *Prevotella* (24%), *Lactobacillus* (10%), *Clostridium* (8%), and *Treponema* (7%).

### 3.2. Fecal Microbiota Diversity Correlated with Feeding Condition and Time Point

To investigate if the pig fecal microbiota diversity was affected by the experimental diet, the alpha diversity was estimated. Regarding the feeding conditions, the Observed species index (Figure 1A) did not differ between the control and the experimental group, and similar results were also observed for the Chao 1 and Shannon indices (Figure 1B,C), respectively. In addition, no differences were observed between the three time points (Figure 1D–F). Regarding the beta diversity, the Bray-Curtis distance indicated that there was no evident dissimilarity among the samples (Appendix A). Furthermore, the beta diversity metric, represented by PCoA (Figure 2), did not highlight any correlation between the groups according to feeding regime and time point.

### 3.3. Analysis of Microbial Genera Abundance at Different Time Points

The differential abundance analysis was performed to evaluate changes in genus-level abundance in pig fecal microbiota. Significant differences were observed between the three time points considered. As can be observed in Table 2, several genera were differentially represented over time, with *Oxalobacter* and *Parasutterella* as the genera most positively modulated by the experimental diet. Conversely, *Duodenibacillus* and *Mogibacterium* were among the most negatively regulated. Interestingly, *Corynebacterium* and *Mogibacterium* genera, known for including pathogenic species, were less abundant. 

## 4. Discussion

The present study investigated the potential influence of an experimental liquid whey-supplemented diet on the fecal bacterial community in pigs belonging to the Nero Siciliano breed. To our knowledge, this work is the first report describing the fecal microbiota of the Nero Siciliano pigs indoor-reared on a commercial farm, thus providing new information about this topic. At the time of writing, no data were available in the scientific literature concerning the potential effect of a liquid whey-supplemented diet on the fecal microbiota of this autochthonous pig breed. Furthermore, the literature does not provide sufficient information about the use of LW in pigs as a co-feed, especially in autochthonous breed feeding. The chemical and microbiological composition of LW can be affected by several factors, including the management of cheesemaking processes [53]. Notwithstanding, a large part of the diverse and complex microbiota of milk can remain in LW after milk treatment (e.g., health-promoting bacteria such as *Lactobacilli* and *Bifidobacteria*) [53]. 

The 16S rRNA gene sequencing approach allowed us to identify a total of 856 OTUs at the genus level, indicating that the feces of this pig breed are populated by a complex community of bacteria, providing additional insights into the ecosystem structure of the rectal section. A previous characterization, based on the whole-metagenome shotgun sequencing of the fecal microbiome associated with Nero Siciliano pig breed, has been reported by Giuffrè et al. [54]. Taxonomically, the results reported in this study, although providing more details regarding the composition of the bacterial community, are globally comparable with those previously reported in 2021 [54]. However, our statistical analysis suggested that the combination of LW in the diet did not affect the alpha diversity metrics of the fecal microbiota. In particular, the Wilcoxon and Kruskal-Wallis values indicated no significant differences in the bacterial composition in terms of richness and evenness between the feeding condition and the three time points (Figure 1). This was also supported by beta diversity (Appendix A) and PCoA analysis (Figure 2). Similar results were obtained by Miragoli et al. [55] when evaluating the effect of other co-feed supplementation in crossbred pigs. From a biological point of view, these findings could be explained by the fact that different microbial communities are accountable for maintaining stability and resistance, even in the face of disturbances and environmental changes [56]. 

In this study, the collection of fecal samples at different time points led us to ask if there might be a potential effect of the experimental diet on the fecal microbiota over time. In fact, data obtained from the abundance analysis at different time points supported our assumption, as twenty bacterial genera were significantly different and abundant (Table 2). Among these, the genera *Bifidobacterium*, *Cellulosilyticum*, *Lachnotalea*, *Ruminococcus*, *Saccharofermentans*, *Phascolarctobacterium*, *Oxalobacter*, and *Parasutterella* were positively modulated. Several species within some of these genera (e.g., *Bifidobacterium*, *Ruminococcus*, and *Oxalobacter*) are often employed as probiotics and in the production of feed additives to prevent diarrhea or to improve growth in mammals [57,58]. Therefore, the use of a co-feed as LW could potentially improve the pig fecal microbiota by increasing beneficial bacteria and reducing the number of potential harmful taxa. Furthermore, the remarkable abundance of the genus *Cellulosilyticum*, known as a symbiont involved in the degradation of dietary fiber in the late stage of swine growth, could be helpful in favoring digestive intestinal processes [59].

Another interesting result of our study was the low detection of bacteria belonging to *Corynebacterium* and *Mogibacterium* genera that are usually associated with pathogenic states. These genera were found to be less abundant over time, thus suggesting that the co-feed diet possibly affected their presence in the feces. It should be noted that some species belonging to the *Corynebacterium* genus (i.e., *C. diptherium*), a potential opportunistic pathogen, have been reported to be associated with intestinal disease in humans and pigs [60,61,62]. In addition, the *Mogibacterium* genus was shown to be increased in mucosa-associated microbiota of patients affected by colon cancer [63] and to be decreased in the stool of neonatal pigs fed with beneficial prebiotic formulations [64]. Furthermore, a high relative abundance of this genus was detected in both mucosal scrapings and luminal samples from pigs with swine dysentery [65]. However, on the basis of this result, we can hypothesize that integration of LW affected the abundance of these genera and therefore further studies, including those using proteomics [66], could be helpful to identify potential biomarkers or microbial signatures for use in clinical practice [67].

## 5. Conclusions

Our study presents the first report exploring the composition of the fecal bacterial community of the autochthonous pig breed Nero Siciliano fed with an alternative by-product. Although supplementation with the dairy by-product co-feed did not affect the microbial diversity, its integration over time contributed to modulating the abundance of several beneficial genera of the bacterial community. Therefore, based on the results obtained in this study, we can assume that a diet supplemented with a low-cost dairy by-product co-feed such as liquid whey might be routinely employed in future in swine breeding to improve animal health.

## Figures and Tables

**Figure 1 animals-13-00642-f001:**
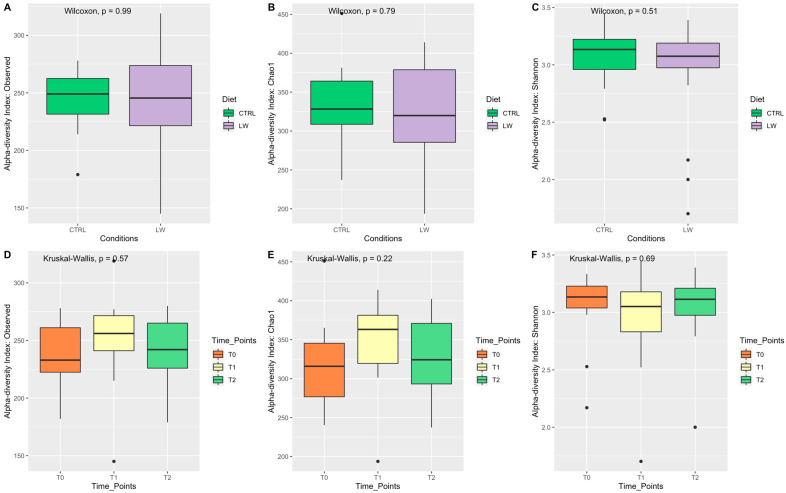
Box plots of the alpha diversity index. Each plot represents the interquartile range while the line that splits the box into two parts represents the median. Plots are graphically presented according to feeding condition and time point.

**Figure 2 animals-13-00642-f002:**
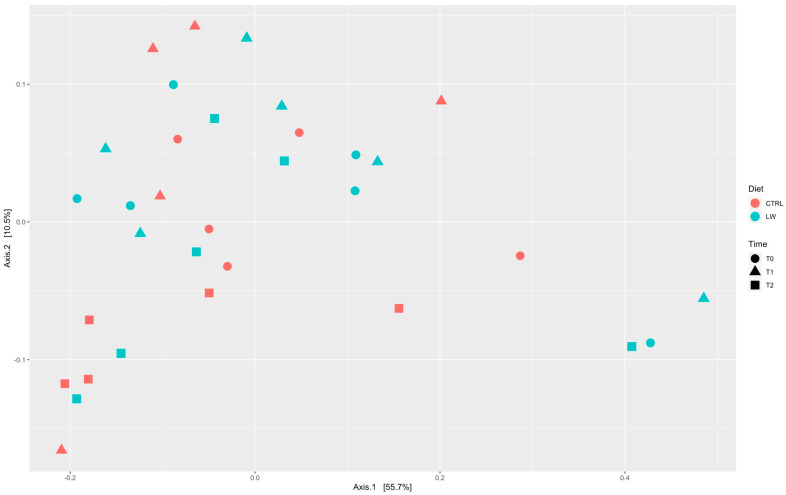
PCoA plot based on Bray-Curtis distances of the fecal microbial communities of Nero Siciliano pigs.

**Table 1 animals-13-00642-t001:** Ingredients and nutritional composition of the experimental diet.

Ingredients	g/Kg of DM
Corn	550
Broad bean	125
Peas beans	110
Sunflower meal (38% CP)	80
Wheat middling	70
Carob	30
Sugar cane molasses	13
**Analytical components ^1^**	
CP	17.4
CFa	5.7
CFi	4.5
Ash	5.3
Calcium	0.6
Phosphorus	0.5
Sodium	0.2
Lysine	1.2
Methionine	0.4
**Additive components**	
Vitamin B1	1.0 mg
Vitamin B2	3.0 mg
Vitamin B6	1.5 mg
Vitamin B12	0.015 mg
Vitamin D3	(1.000 UI)
Vitamin E	20 mg
Vitamin K3	1.0 mg
Niacin	15.0 mg
Calcium-D	10.3 mg
Choline	200 mg
Cu	14.0 mg
Fe	89.8 mg
I	0.50 mg
Mn	39.9 mg
Se	0.15 mg
Zn	99.6 mg
Biotin	0.10 mg
DL-Methionine	0.12 mg
Lysine	500 mg

^1^ % on a wet basis; DM = dry matter; CP = crude protein; CFa = crude fat; CFi = crude fiber; Data reported were provided by the commercial farm.

**Table 2 animals-13-00642-t002:** Overview of the differential abundance analysis reporting the significant variation at genus level in the time periods considered in the trial.

Phylum	Family	Genus	Log2 Fold Change *	*p*-Value	*p*-Adj
*Actinobacteria*	*Bifidobacteriaceae*	*Bifidobacterium*	1.79	2.78 × 10^−4^	4.93 × 10^−3^
	*Corynebacteriaceae*	*Corynebacterium*	−1.35	4.81 × 10^−3^	4.02 × 10^−2^
	*Atopobiaceae*	*Olsenella*	−2.34	1.21 × 10^−4^	2.87 × 10^−3^
	*Eggerthellaceae*	*Slackia*	−1.78	2.23 × 10^−3^	2.43 × 10^−2^
*Firmicutes*	*Lactobacillaceae*	*Lactobacillus*	−1.11	6.28 × 10^−3^	4.69 × 10^−2^
	*Clostridiales Family XIII. Incertae Sedis*	*Mogibacterium*	−2.45	7.37 × 10^−7^	5.24 × 10^−5^
	*Lachnospiraceae*	*Blautia*	−0.97	6.66 × 10^−3^	4.73 × 10^−2^
		*Cellulosilyticum*	1.97	3.74 × 10^−3^	3.31 × 10^−2^
		*Coprococcus*	−2.36	2.00 × 10^−4^	4.06 × 10^−3^
		*Lachnospira*	−1.62	2.64 × 10^−3^	2.68 × 10^−2^
		*Lachnotalea*	1.79	1.63 × 10^−3^	2.19 × 10^−2^
	*Ruminococcaceae*	*Anaeromassilibacillus*	−2.05	9.01 × 10^−4^	1.42 × 10^−2^
		*Harryflintia*	0.07	6.77 × 10^−8^	9.62 × 10^−6^
		*Ruminococcus*	1.41	4.77 × 10^−5^	1.35 × 10^−3^
		*Saccharofermentans*	2.24	1.87 × 10^−3^	2.22 × 10^−2^
	*Acidaminococcaceae*	*Phascolarctobacterium*	1.62	3.78 × 10^−5^	1.34 × 10^−3^
*Proteobacteria*	*Oxalobacteraceae*	*Oxalobacter*	5.87	1.69 × 10^−3^	2.19 × 10^−2^
	*Sutterellaceae*	*Duodenibacillus*	−2.64	2.94 × 10^−3^	2.78 × 10^−2^
		*Parasutterella*	3.99	1.31 × 10^−6^	6.19 × 10^−5^
	*Pasteurellaceae*	*Actinobacillus*	−0.22	5.18 × 10^−3^	4.09 × 10^−2^

* Changes in genus-level abundance refer to time points T2 vs. T0.

## Data Availability

Data supporting the results of this study have been deposited into the Sequence Read Archive database under the following study accession number SUB12344409 associated with the BioProject ID PRJNA909983 and in Figshare under the project entitled “Characterization of the Nero Siciliano pig fecal microbiota after a liquid whey-supplemented diet” at https://doi.org/10.6084/m9.figshare.21802380.v1 (accessed on 3 January 2023).

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
