# Peer review of "Characterization of the Nero Siciliano Pig Fecal Microbiota after a Liquid Whey-Supplemented Diet"

_animals, 2023, doi:10.3390/ani13040642_

Round 1

Reviewer 1 Report

This case-control study aimed to investigate the fecal microbiota of the Nero Siciliano pig after a liquid whey co-feed supplemented diet using a  metagenomic approach. The topics is of interest and I would like to congratulate with Authors for their effort

My comments are:

1)   Add in the discussion section a word on the importance of animal serum proteomic in order to identify novel biomarkers for animal welfare, early diagnosis, prognosis and monitoring of infectious disease and treatments.

[Di Girolamo, Amato AD', Lante I, Signore F, Muraca M, Putignani L. Farm animal

299 serum proteomics and impact on human health. Internat J Molec Sci 2014;15, 15396-15411. 300 doi:10.3390/jims150915396.]

2)   The role of gut microbiota both in human health and in diseases is the subject of intense investigation. Add this important role in the introduction (lines 85-86)[Muraca M, putignani L, Fierabracci A, Teti A, Perilongo G. Gut microbiota-derived outer membrane vesicles: under-recognized major players in health and disease? Discovery Medicine; 2015. 19(106):343-348].

3)   Add a word in the discussion section about the importance of proteomics and the application of Matrix Assisted Laser Desorption Ionization Time of Flight (MALDI-TOF) MS as a suitable platform for providing a high-throughput support to clinics [Greco V, Piras C, Pieroni L, Ronci M, Putignani L, Roncada P, Urbani A. Applications of MALDI-TOF mass spectrometry in clinical proteomics. Expert Rev Proteomics. 2018;15(8):683-696.]

Author Response

Reviewer 1

This case-control study aimed to investigate the fecal microbiota of the Nero Siciliano pig after a liquid whey co-feed supplemented diet using a metagenomic approach. The topics is of interest and I would like to congratulate with Authors for their effort.
My comments are:

1) Add in the discussion section a word on the importance of animal serum proteomic in order to identify novel biomarkers for animal welfare, early diagnosis, prognosis and monitoring of infectious disease and treatments. [Di Girolamo, Amato AD', Lante I, Signore F, Muraca M, Putignani L. Farm animal serum proteomics and impact on human health. Internat J Molec Sci 2014, 15, 15396–15411. doi:10.3390/jims150915396.]

Response 1:

We thank the reviewer for his/her comments. As recommended, we integrated the suggested citation in the discussion section of our manuscript.

2) The role of gut microbiota both in human health and in diseases is the subject of intense investigation. Add this important role in the introduction (lines 85-86) [Muraca M, putignani L, Fierabracci A, Teti A, Perilongo G. Gut microbiota-derived outer membrane vesicles: under- recognized major players in health and disease? Discovery Medicine 2015, 19(106):343–348].

Response 2:

We thank the reviewer for his/her comments. As recommended, we integrated the suggested citation in the introduction of our manuscript.

3) Add a word in the discussion section about the importance of proteomics and the application of Matrix Assisted Laser Desorption Ionization Time of Flight (MALDI-TOF) MS as a suitable platform for providing a high-throughput support to clinics [Greco V, Piras C, Pieroni L, Ronci M, Putignani L, Roncada P, Urbani A. Applications of MALDI-TOF mass spectrometry in clinical proteomics. Expert Rev Proteomics 2018, 15(8):683–696.]

Response 3:

We thank the reviewer for his/her comments. As recommended, we integrated the suggested citation in the discussion section of our manuscript.

Reviewer 2 Report

My comments for this manuscript below:

- Please add growth performance data;

- Please insert the Table of ingredient diet formulations (but not in the supplemental information); and

- Line 105: the initial body weight of 19 kg can not be right to at the age of 28 days for their study pigs? please check and revise this.

Author Response

Reviewer 2

My comments for this manuscript below:

- Please add growth performance data;

Response 1:

The authors thank the reviewer for his/her helpful comments. As required by the reviewer, we updated the results with the growth performance data in the results section.

- Please insert the Table of ingredient diet formulations (but not in the supplemental information);

Response 2:

We thank the reviewer for the suggestion. The table was added within the revised version of the manuscript.

- Line 105: the initial body weight of 19 kg cannot be right to at the age of 28 days for their study pigs? please check and revise this.

Response 3:

Thank you very much for this precious remark. We apologize for our inaccuracy but a mistake occurred during the initial text editing. The initial body weight of pigs was referred to the age of 58 days and not 28. Therefore, we revised this detail in the text.